# Impact of Diagnosis-Intervention Packet (DIP) reforms on inpatient services for low-income populations in central China: A multi-stage interrupted time-series analysis

Keyi Shen, Yingying Tao, Yile Li, Ziqian Jin, Chengcheng Li, Dan Wu, Xuehui Meng*

School of Humanities and Management, Zhejiang Chinese Medical University, Hangzhou, China.

* mengxuehui@aliyun.com

## Abstract

As global healthcare costs continue to rise, concerns about health equity have become increasingly prominent. In response, China introduced the Diagnosis-Intervention Packet (DIP) reform in 2021 to optimize healthcare resource allocation and control costs. While the reform has been widely discussed in terms of its overall cost-control effects, its heterogeneous impact on low-income populations, especially across different hospital tiers, remains unclear. This study aims to fill this gap by examining the differentiated impact of the DIP reform on low-income patients' inpatient service utilization. Using multi-stage interrupted time series (ITS) analysis, we analyzed 1.17 million hospitalization records from low-income patients in City S, a pilot city in central China. The study reveals that DIP significantly reduced total hospitalization costs and length of stay (LOS) but led to increased readmission rates, indicating a trade-off between efficiency gains and potential risks to care quality. The reform's effects varied by hospital tier: primary hospitals saw increased demand for non-acute hospitalization due to reduced out-of-pocket (OOP) payments, exposing resource shortages; secondary hospitals balanced cost control and revenue by shortening stays and increasing admission frequency, which raised readmission risks; and tertiary hospitals, treating critically ill patients, enhanced treatment completeness, though multiple hospitalizations were still needed for full recovery. The study introduces a two-dimensional framework— "hospital tier-policy cycle"—demonstrating that differences in service capacity across hospital levels are central to the heterogeneous effects of the DIP reform. These findings suggest that future policies should strengthen primary care resources, introduce quality assurance mechanisms, and consider bundled payment models for critical care. This research contributes valuable insights for optimizing equity in DIP reform and offers implications for similar healthcare payment systems globally.

**Data availability statement:** The data that support the findings of this study were obtained from the Hubei Provincial Medical Security Bureau. Due to legal restrictions under China's Personal Information Protection Act (2021) and the Regulations on the Supervision and Administration of Medical Insurance Fund Use (2021), which mandate strict protection of patient privacy and healthcare data security, the raw dataset cannot be made publicly available. Researchers interested in accessing the de-identified data can apply by submitting a formal request to the Hubei Provincial Medical Security Bureau (contact email: [ybjysq@hubei. gov.cn]). Applications must include a detailed research proposal, ethical approval documentation from the applicant's institution, and a signed data use agreement prohibiting redistribution or commercial use. Approved researchers can access the data only within the secure facilities of the platform; external data transfer is prohibited. The authors confirm that they had no special privileges in accessing the data, and all analyses were conducted in compliance with the platform's governance protocols.

**Funding:** The author(s) received no specific funding for this work.

**Competing interests:** The authors have declared that no competing interests exist.

## Introduction

Healthcare systems worldwide are under tremendous pressure due to aging populations, a high prevalence of chronic diseases, and skyrocketing healthcare costs. Policymakers are actively seeking strategies that balance quality, outcomes, and affordability to maintain public health while rationalizing resource use [1]. However, in addition to improving quality and efficiency, effectively controlling expenditures has also become a major challenge for healthcare systems globally [2]. Against this backdrop, healthcare payment reforms have been viewed as an important tool to optimize resource allocation, control costs, and enhance health equity. [3]. Among these, the Diagnosis Related Groups (DRG) payment model has emerged as a central strategy for many countries exploring healthcare payment reforms, due to its ability to curb overutilization and improve resource efficiency [4–6]. While the objectives of implementing DRG-based payment vary across countries, four primary goals are typically identified: controlling costs, enhancing inpatient care efficiency, and improving care quality [7]. By standardizing hospitalization costs based on disease categories, DRG payment models not only incentivize hospitals to improve efficiency but also help alleviate the financial burden on patients [8]. However, the limitations of DRG payment models have become increasingly apparent. For example, the complexity of pricing criteria may encourage hospitals to prioritize low-cost patients, potentially neglecting the healthcare needs of vulnerable groups, such as low-income populations and critically ill patients [9,10]. Additionally, the successful implementation of the DRG payment in many countries relies on advanced healthcare information systems, raising concerns about its applicability in developing regions [11].

Due to the limited applicability of the DRG model in China, there is an urgent need for the country to explore payment methods that align with its own realities [11]. In 2020, the Chinese government developed and implemented a new payment method, the DIP payment model, which utilizes an integrated management system based on big data to explore its potential advantages [12]. The DIP payment model employs this system to identify the common characteristics of "disease diagnosis + treatment modality," objectively classifying case data and establishing standardized categorizations of each disease-treatment combination within a specific regional context. By objectively reflecting disease severity, treatment complexity, resource consumption, and clinical norms, the system enables more precise categorization of over 10,000 patient groups. Using historical cost data, patient groups are assigned relative weights that reflect resource utilization across different groups, which are subsequently translated into payments based on global insurance budgets [13]. Through harmonized disease coding and cost-of-care pricing, efforts are made to lower the implementation threshold while preserving the advantages of the DRG model. Since 2020, the DIP payment model has been implemented in 71 pilot cities nationwide, gradually replacing the fee-for-service (FFS) model. Compared to DRG, the DIP model offers a more refined and flexible classification of disease types, facilitating broader coverage of healthcare service scenarios.

However, while innovating payment methods, some scholars have argued that any payment system may exacerbate the challenges faced by low-income and racial

minority groups [14,15]. Health equity issues, particularly those affecting low-income groups, have long been a focal point in global health. Economic disparities and geographic location are key factors influencing people's access to healthcare services and opportunities for health equity [16]. In China, the challenges related to healthcare utilization and financial burden are especially pronounced among low-income groups [17]. Under the FFS payment, hospitals tend to increase the volume of medical services to generate more revenue, and this profit-driven behavior may further intensify the financial burden on low-income patients [18]. Moreover, high hospitalization costs often discourage low-income populations from seeking medical care, there by worsening their health outcomes [19]. The primary goal of DIP payment reforms is to control healthcare costs, with the intention of alleviating patients' financial burden by standardizing the pricing of medical care. In the context of DIP payment reform, evaluating its impact on healthcare utilization among low-income groups has become a critical issue.

To our knowledge, few studies have focused on the impact of DIP payment reform on low-income populations [20–22]. While some prior research on DIP payment reform has concentrated on its effects on healthcare cost containment and efficiency, it has generally been from the perspective of the overall patient population [13,22]. As the reform has progressed, the scope of research on DIP has gradually expanded. For example, Zhang et al. evaluated the impact of DIP reform on the total cost of hospitalization versus the number of days of hospitalization for women and children in Guangdong Province, China [23]. Tan et al. examined the cost-shifting behavior of hospitals with different grades in treating patients with coronary heart disease in Zunyi City under the DIP reform [24]. Lin et al. revealed differences in physician behavior under FFS versus DIP payment models [25]. Additionally, other scholars have assessed the impact of DIP payments on inpatient admissions to hospitals of different grades and types, drawing mixed conclusions regarding costs, efficiency, and hierarchical diagnosis and treatment [26,27]. However, most of these studies have focused on western and southern cities in China, with a lack of exploration of central cities, and even fewer studies have addressed issues of fairness.

Methodologically, most current studies employ a single-stage ITS analysis, which assumes that the effects of policy changes are immediately apparent and constant [13,26–29]. This assumption is difficult to maintain in complex policy contexts, particularly when a policy like the DIP payment reform is implemented incrementally, potentially exhibiting staged effects and variations over time [30]. Furthermore, most existing studies analyze the effects of payment reform from the supply-demand perspective, but they overlook behavioral heterogeneity in a tiered healthcare system [31].

Thus, the dynamic impact of DIP payment reforms on inpatient services for low-income populations over time, particularly the differences between hospital tiers, represents a key area for further research. Based on this, this study aims to address three primary questions: First, does the behavioral response of low-income populations, as a price-sensitive group, exacerbate the unintended effects of payment reforms? Second, does the existing literature, which generally assumes 'homogenization of hospitals,' overlook the spatial differences in strategies employed by hospitals of different tiers under the hierarchical diagnosis and treatment system? Third, is it possible to capture the dynamics of the mechanisms driving the evolution of the effects of the multi-stage DIP policy? This study seeks to overcome these limitations by constructing a two-dimensional analytical framework, the 'hospital tier-policy cycle.

## Materials and methods

### Research background and design

We selected S City in Hubei Province, located in central China, as the research subject. In 2023, the city achieved a GDP of 235.903 billion RMB, ranking seventh within the province, and had a resident population of 3,153,200, reflecting a medium level of economic and social development. S City has made significant strides in poverty eradication, encompassing six national-level poverty-stricken counties. Between 2016 and 2020, the city successfully lifted 810,000 individuals out of poverty. By 2021, the low-income population in S City was reduced to 214,000 individuals, including recipients of the minimum subsistence guarantee, special hardship cases, orphans, and others, each with an average annual income

of 20,000 RMB or below. Social insurance coverage for this population segment reached 100%. Additionally, the city's healthcare infrastructure consists of a three-tier service network comprising 262 hospitals, including 214 primary hospitals, 44 secondary hospitals, and 6 tertiary hospitals.

At the end of 2020, S City was selected as one of the first 71 pilot cities nationwide for the DIP reform. In January 2021, the local healthcare security administration officially initiated the DIP payment method reform in S City. After a year of preparation, the DIP payment model was implemented across all medical institutions in the city by January 2022, covering inpatient services. Consequently, we define the period from January 2021 to December 2021 as the pilot phase and from January 2022 to December 2023 as the advancement phase. To control for the impact of other policies, we reviewed relevant literature and policies to ensure that no confounding factors could influence hospital inpatient service behaviors. Methodologically, traditional single-stage ITS assumes that the effect of policy intervention on the outcome variable is singular, immediate, and sustained [32]. However, in practice, the impact of many policies is not immediate, especially in complex systems, and policy effects may vary in pattern and intensity over time [30]. In such cases, multistage interrupted time series analysis can better capture the dynamic effects of policy implementation. Therefore, January 2021 and January 2022 were used as intervention points in this study, employing a multistage interrupted time series design to assess the sustained impact of DIP payment reforms on inpatient services for low-income populations across different hospital tiers in S City. This study aims to comprehensively characterize the changes and sustained impacts of the policy over different time periods.

## Data source and sample

Since S City did not experience a large-scale outbreak of the COVID-19 pandemic until 2022, the population's healthcare needs were adequately met during this period. This study employs claims reimbursement data from the Medical Protection Bureau for low-income populations in S City, Hubei Province, to conduct empirical analysis. The dataset includes 1,207,953 hospitalization claims records for the low-income population in S City, spanning from January 1, 2019, to December 31, 2023. These records provide information on age, gender, admission and discharge dates, type of medical coverage, hospitalization costs, out-of-pocket (OOP) payments, diagnosis, and the level of the hospital visited.

The data were accessed on June 10, 2024, from the Hubei Provincial Healthcare Security Information Platform. Each patient was assigned a unique identification code to ensure privacy, and the dataset was fully anonymized before access. The authors had no access to information that could identify individual participants during or after data collection.

To ensure the accuracy of the analysis, we excluded cases with unknown gender, atypical hospitalization durations, and those not subject to DIP payment criteria (e.g., mental illness and rehabilitation cases). Additionally, outliers and erroneous data were removed to prevent distortion of subsequent analyses. Thus, the final sample size analyzed consisted of 1,172,145 cases. The study was approved by the ethics committee of the authors' university.

## Measurement variables

During the study observation period, we assessed hospital inpatient service utilization among low-income patients using several variables: total hospitalization cost, LOS, OOP ratios, and 30-day readmission rates. To account for inflation, all cost data were adjusted using China's annual consumer price index, with 2019 as the base year. Additionally, we applied the natural logarithm (ln) to the total hospitalization cost per inpatient to normalize its distribution. The LOS was calculated as the difference between the patient's admission and discharge dates, and the OOP ratios were determined by the ratio of costs paid by the patient to the total hospitalization costs. 30-day readmission rates were defined as any rehospitalization within 30 days of discharge for health-related reasons. To ensure the accuracy of this variable, we excluded readmissions due to non-medical factors, such as transfers.

## Statistical analysis

Continuous variables are presented as means (standard deviations), and categorical variables as percentages. We compared patient characteristics and outcome variables before and after the implementation of the DIP reform in S City using t-tests and chi-square tests. This paper employs multi-stage ITS analysis, chosen for its robustness in assessing policy effects in a quasi-experimental setting. It captures both immediate changes and post-intervention trends at different stages, while also accounting for pre-existing trends to isolate the impact of DIP reforms. This method is particularly well-suited for analyzing the results of interventions that are either time-limited or continuously evolving, such as the implementation of DIP payments [32]. Additionally, the decision to use multi-stage ITS instead of a standard ITS stems from the systemic gradualism inherent in DIP reforms [30]. Specifically, there is a difference in policy intensity between the pilot period (2021) and the advancement period (2022), marked by an increase in the standardized coverage of payments. By setting two breakpoints, this approach allows for the capture of hospitals' behavioral transition from 'tentative adaptation' to 'strategic optimization.'

In this study, 60 monthly time points were used as the unit of analysis, with January 2021 and January 2022 designated as the time points for the policy intervention. This design enables the assessment of the impact of the DIP policy on total hospitalization costs, LOS, OOP ratios, and 30-day readmission rates for low-income patients. The model assumes that the DIP policy intervention will affect inpatient service post-intervention. Two ITS regression models were constructed as follows:

$$Y_t = \beta_0 + \beta_1 time + \beta_2 intervention + \beta_3 post + \varepsilon_t \tag{1}$$

$Y_t$ represents the outcome variable at time t, including key outcomes such as total hospitalization costs, LOS, OOP ratios, and others. "Time" denotes the time variable ranging from the start to the end of the study period. The variable "intervention" indicates whether the policy intervention has occurred, taking the value of 0 before the intervention and 1 after the intervention. "Post" refers to the post-intervention time series, and $\varepsilon_t$ is the random error term. $\beta_0$ represents the baseline level, indicating the initial value or baseline state of the outcome variable in the absence of the intervention. $\beta_1$ represents the monthly slope of the outcome variable before the DIP policy intervention. $\beta_2$ captures the difference between the observed outcome at the time of the intervention and the counterfactual outcome, assuming no intervention had occurred. In other words, $\beta_2$ represents the instantaneous change in the outcome variable following the DIP policy intervention. $\beta_3$ reflects the change in the trend of the outcome variable post-intervention compared to the pre-intervention period. Finally, $\beta_1 + \beta_3$ indicates the immediate post-intervention trend.

$$Y_t = \beta_0 + \beta_1 time + \beta_2 intervention + \beta_3 post + \beta_4 intervention_2 + \beta_5 secondtime + \varepsilon_t \tag{2}$$

The definitions of "time," "intervention," and "post," along with the interpretations of $\beta_0$ through $\beta_3$, remain consistent with those in Model (1). The variable "intervention2" indicates whether a second policy intervention occurs specifically during the advancement period of the DIP payment reform, taking the value of 0 before the intervention and 1 after. $\beta_4$ represents the immediate change in the level following the second intervention, while $\beta_5$ captures the change in the slope before and after the second intervention, reflecting the policy's impact relative to the pilot period.

To examine the final effect of the policy and address the limitation that the multi-stage model cannot compare trend changes between the pre-reform and advancement periods, we employ Model (1) to analyze these changes. Specifically, we regress data from the two-year (24-month) pre-reform period against data from the advancement period. The results are represented by $\beta_6$ (the instantaneous change in the advancement period compared to the pre-reform period) and $\beta_7$ (the change in the slope of the advancement period compared to the pre-reform period).

To account for model autocorrelation, we initially apply the Newey-West method with zero lags, thereby fitting an Ordinary Least Squares (OLS) model. Subsequently, we assess the level of autocorrelation in the model using the act-est command. If first-order autocorrelation is detected, we employ the Prais-Winsten method to estimate the regression. The adjusted Durbin-Watson (DW) statistic is then utilized to evaluate the effectiveness of the model's autocorrelation adjustment. Since the Prais-Winsten method is limited to first-order autocorrelation, we apply the Newey-West method, specifying an appropriate number of lags to address autocorrelation when it is absent or when second order and higher autocorrelations are present.

In the sensitivity analysis, we evaluate whether changes in the outcome variables can be attributed to the implementation of the DIP policy by establishing pseudo time points during the non-intervention period [33]. To further validate the robustness of the model results and ensure that the observed policy effects originate from the implementation of the DIP payment reforms, we conduct a pseudo point-of-intervention test. This involves selecting two time periods before and after the DIP policy intervention and establishing two dummy policy intervention points to determine whether the outcome variables would have changed significantly in the absence of actual policy interventions. In this study, we designate June 2020 and June 2021 as pseudo-intervention points to simulate the implementation of the DIP payment reforms at these specific times and compare them to the actual intervention points (January 2021 and January 2022). This setup enables us to test whether the model would exhibit similarly significant changes in the absence of actual policy interventions (S1 Table). All tests with $P<0.05$ were considered statistically significant. All statistical analyses were conducted using Stata 17.0 for Windows.

### Ethical statement

The data for this study were obtained from the Hubei Provincial Healthcare Security Information Platform, covering hospitalization claims records for low-income populations in S City, Hubei Province, from January 1, 2019, to December 31, 2023. Patients' information was fully anonymized, with only unique identifiers retained for analysis purposes. The data were accessed on June 10, 2024, after obtaining approval from the Ethics Committee of Zhejiang University of Traditional Chinese Medicine (Approval Number: 20240516–3). The authors had no access to any personally identifiable information during or after the study. The Bureau of Medical Security authorized the use of the anonymized dataset for research purposes. The requirement for individual informed consent was waived due to the retrospective nature of the study. This study adhered strictly to the ethical principles outlined in the Declaration of Helsinki, and all data processing methods complied with relevant legal and ethical standards.

### Informed Consent Statement

Informed consent is not required because the Medicare billing information database of the Bureau of Medical Security has anonymized each patient, using only the personal number as an identifying sentence, and does not contain any personal information that may be associated with an individual participant.

## Results

### Descriptive analysis

Table 1 presents the between-group comparisons of patients' basic characteristics and hospitalization service variables before and after the DIP payment reform. The average age of patients increased from 56.51 years (2019–2020) before the reform to 59.31 years (2021–2023) after the reform. Regarding gender distribution, the proportion of male patients slightly rose from 56.60% before the reform to 58.55% after the reform. Health insurance coverage remained at 100%, with an increase in the number of individuals enrolled in basic health insurance for urban workers following the reform. The hierarchical distribution of hospitals attended by the low-income population shifted, with the proportion of patients

**Table 1. characteristics of a sample of low-income hospitalized patients in S. 2019–2023.**

| Variables | Before DIP reform | | | After DIP reform | | | | p value |
|---|---|---|---|---|---|---|---|---|
| | Year | | Total | Year | | | Total | |
| | 2019 | 2020 | | 2021 | 2022 | 2023 | | |
| Discharge cases,No. | 234772 | 295146 | 529918 | 310747 | 153410 | 178070 | 642167 | |
| Age,mean(SD) | 56.53(18.03) | 56.49(18.67) | 56.51(18.47) | 56.47(19.49) | 61.76(16.15) | 62.16(16.28) | 59.31(18.01) | 0.000 |
| sex,No.(%) | | | | | | | | 0.000 |
| Male | 132888(56.60) | 161858(54.84) | 294746(55.62) | 169387(54.51) | 94166(61.38) | 112491(63.17) | 376044(58.55) | |
| Female | 101884(43.40) | 133288(45.16) | 235172(44.38) | 141360(45.49) | 59244(38.62) | 65579(36.83) | 266183(41.45) | |
| medical insurance,No.(%) | | | | | | | | 0.000 |
| Urban Resident Basic Medical Insurance | 234772(100.00) | 295146(100.00) | 529918(100.00) | 310298(99.86) | 152707(99.54) | 177145(99.48) | 640150(99.68) | |
| Urban Employee Basic Medical Insurance | 0(0.000) | 0(0.00) | 0(0.00) | 449(0.14) | 704(0.46) | 925(0.52) | 2078(0.32) | |
| Hospital level,No.(%) | | | | | | | | 0.000 |
| Tertiary(N=6) | 32305(13.76) | 39161(13.27) | 71466(13.49) | 46758(15.05) | 17848(11.63) | 20125(11.30) | 84731(13.19) | |
| Secondary(N=44) | 82886(35.30) | 107882(36.55) | 190768(36.00) | 115404(37.14) | 53117(34.62) | 56196(31.56) | 224717(34.99) | |
| Primary(N=214) | 119581(50.93) | 148103(50.18) | 267684(50.51) | 148585(47.82) | 82445(53.74) | 101749(57.14) | 332779(51.82) | |
| Total hospitalization costs, mean(SD),RMB | 5371.93(10848.85) | 5330.72(10207.44) | 5348.97(10496.45) | 5463.20(10705.77) | 4970.97(9440.55) | 4687.01(8486.57) | 5130.40(9840.19) | 0.000 |
| Length of stay,mean(SD),day | 10.29(11.55) | 9.15(7.02) | 9.65(9.32) | 8.74(6.64) | 8.84(6.06) | 8.47(5.20) | 8.69(6.13) | 0.000 |
| 30-Day Readmission, No.(%) | 37065(15.79) | 39415(13.35) | 76480(14.43) | 41584(13.38) | 23560(15.36) | 31017(17.42) | 96161(14.97) | 0.000 |
| Out-of-pocket ratios, mean(SD), (%) | 9.75(7.84) | 12.69(7.85) | 11.39(7.98) | 13.45(7.98) | 12.15(6.77) | 10.11(7.02) | 12.21(7.58) | 0.000 |

attending first-class hospitals increasing from 50.51% before the reform to 51.82% after the reform, while the proportions attending secondary and tertiary hospitals declined. In terms of total hospitalization costs, the average cost per patient decreased significantly from 5,371.93 RMB before the reform to 5,130.40 RMB after the reform. The number of LOS also decreased from an average of 10.29 days before the reform to 8.69 days after the reform. Conversely, the OOP ratios increased from 11.39% to 12.21%.

## Total hospitalization costs

Table 2 and Fig 1 show that total hospitalization costs increase with hospital level. Additionally, the trends in these costs varied across different hospital levels and stages of the DIP policy intervention. In tertiary hospitals, the trend showed a decrease of 1.9% per month before the reform ($\beta_1 = -0.019$, $P < 0.001$), followed by an increase of 3.3% per month during the policy transition ($\beta_3 = 0.052$, $P < 0.001$), and then a decrease of 0.2% per month after the policy advancement ($\beta_5 = -0.035$, $P < 0.001$). In contrast, the trends in primary hospitals were completely opposite to those in tertiary hospitals.

## Out-of-pocket ratios

Tertiary hospitals had the highest OOP ratios for low-income patients, while primary hospitals had the lowest (Table 2, Fig 2). The trends in OOP ratios followed similar patterns across different hospital levels and stages of the DIP policy intervention. Before the policy intervention, the OOP ratios for hospitalized patients increased at all hospital levels, with the slowest increase observed in primary hospitals ($\beta_1 = 0.266$, $P < 0.001$). Tertiary hospitals saw the highest monthly increase in OOP ratios ($\beta_1 = 0.464$, $P < 0.001$). During the policy transition period, the OOP ratios continued to rise, though the rate of increase slowed. In the policy advancement period, OOP ratios significantly decreased across all hospital levels. Overall, compared to the pre-reform period, the policy intervention led to a reduction in OOP ratios, alleviating the financial burden on patients.

## 30-Day readmission rates

In the pre-reform period, 30-day readmission rates for low-income patients decreased significantly at all hospital levels, with tertiary hospitals showing the largest decline at -0.369% per month ($\beta_1 = -0.369$, $P < 0.001$) (Table 2, Fig 3). During the policy transition period, readmission rates increased significantly in both secondary hospitals (0.202% per month, $\beta_3 = 0.472$, $P < 0.001$) and tertiary hospitals (0.316% per month, $\beta_3 = 0.685$, $P < 0.001$). In the policy advancement period, the increase in readmission rates in tertiary hospitals slowed, though this change was not statistically significant ($\beta_5 = -0.136$, $P = 0.492$). Conversely, primary hospitals experienced an acceleration in readmission rates, with an increase of 0.206% per month ($\beta_5 = 0.189$, $P = 0.026$). Overall, readmission rates for hospitalized patients increased across all hospitals during the advancement period compared to the pre-reform period.

## Length of stay

In the pre-reform period, both secondary and tertiary hospitals experienced significant decreases in the LOS, with a decline of -0.140 days per month ($\beta_1 = -0.140$, $P < 0.001$) in secondary hospitals and -0.137 days per month ($\beta_1 = -0.137$, $P < 0.001$) in tertiary hospitals (Table 2, Fig 4). During the policy transition period, the LOS in tertiary hospitals rebounded, increasing at a rate of 0.073 days per month ($\beta_3 = 0.210$, $P < 0.001$), while the decline in secondary hospitals slowed to -0.018 days per month ($\beta_3 = 0.122$, $P = 0.020$). In the policy advancement period, the trend in tertiary hospitals reversed again, with a significant decrease of -0.034 days per month ($\beta_5 = -0.107$, $P = 0.005$). Overall, during the advancement period, the LOS decreased across all hospital levels compared to the pre-reform period, though the decrease slowed in secondary and tertiary hospitals.

**Table 2. Results of multi-stage interruption time series analysis.**

| | β₁ Estimate (95%CI) | β₂ Estimate (95%CI) | β₃ Estimate (95%CI) | β₄ Estimate (95%CI) | β₅ Estimate (95%CI) | β₆ Estimate (95%CI) | β₇ Estimate (95%CI) | β₀ Estimate (95%CI) |
|---|---|---|---|---|---|---|---|---|
| **Total hospitalization costs** | | | | | | | | |
| Total hospitals | 0.002 (-0.004,0.009) | 0.072 (-0.023,0.167) | -0.015 (-0.023,-0.007)*** | 0.095 (0.049,0.140)*** | 0.009 (0.004,0.014)*** | 0.015 (-0.058,0.087) | -0.006 (-0.011,-0.001)* | 7.985 (7.892,8.078)*** |
| Tertiary hospitals | -0.019 (-0.028,-0.009)*** | -0.041 (-0.213,-0.131) | 0.052 (0.026,0.077)*** | -0.097 (-0.278,-0.084) | -0.035 (-0.061,-0.009)** | 0.246 (0.101,0.391)** | 0.015 (0.003,0.027)* | 9.267 (9.137,9.397)*** |
| Secondary hospitals | 0.005 (-0.001,0.009) | 0.011 (-0.044,0.065) | -0.014 (-0.020,-0.009)*** | 0.091 (0.047,0.135)*** | 0.007 (0.004,0.011)*** | -0.016 (-0.083,0.051) | -0.007 (-0.012,-0.002)* | 8.445 (8.366,8.524)*** |
| Primary hospitals | 0.009 (0.005,0.013)*** | -0.012 (-0.078,0.054) | -0.025 (-0.032,-0.017)*** | 0.161 (0.099,0.224)*** | 0.020 (0.013,0.027)*** | -0.097 (-0.0177,-0.017)* | -0.005 (-0.013,0.004) | 7.285 (7.224,7.346)*** |
| **Out-of-pocket ratios** | | | | | | | | |
| Total hospitals | 0.335 (0.215,0.455)*** | -1.550 (-3.039,-0.061)* | -0.320 (-0.450,-0.190)*** | -0.099 (-0.671,0.473) | -0.216 (-0.281,-0.151)*** | -1.472 (-2.988,0.045) | -0.536 (-0.663,-0.409)*** | 6.896 (4.895,8.897)*** |
| Tertiary hospitals | 0.464 (0.287,0.641)*** | -2.519 (-4.695,-0.344)* | -0.403 (-0.637,-0.169)** | 1.134 (-0.427,2.696) | -0.265 (-0.438,-0.092)** | 1.344 (-1.645,4.332) | -0.688 (-1.111,-0.266)** | 11.361 (8.262,14.461)*** |
| Secondary hospitals | 0.387( 0.229,0.544)*** | -2.450 (-4.532,-0.368)* | -0.354 (-0.522,-0.187)*** | 0.392 (-0.462,1.246) | -0.243 (-0.327,-0.159)*** | -0.003 (-2.380,2.376) | -0.61- (-0.941,-0.279)*** | 9.229 (6.787,11.670)*** |
| Primary hospitals | 0.266 (0.167,0.365)*** | -1.526 (-2.818,-0.233)* | -0.223 (-0.326,-0.119)*** | -0.384 (-0.758,-0.010)* | -0.185 (-0.231,-0.139)*** | -0.477 (-2.107,1.154) | -0.407 (-0.618,-0.195)*** | 4.051 (2.408,5.694)*** |
| **30-Day readmission rates** | | | | | | | | |
| Total hospitals | -0.236 (-0.328,-0.144)*** | 0.818 (-0.839,2.476) | 0.347 (0.137,0.557)** | -0.013 (-1.946,1.920) | 0.087 (-0.114,0.289) | 2.132 (0.192,4.072)* | 0.434 (0.307,0.562)*** | 17.516 (16.230,18.802)*** |
| Tertiary hospitals | -0.369 (-0.481,-0.256)*** | 1.067 (-1.468,3.601) | 0.685 (0.325,1.045)*** | -0.101 (-3.986,3.785) | -0.136 (-0.532,0.259) | 4.424 (0.248,8.599)* | 0.546 (0.227,0.865)** | 26.082 (24.649,27.515)*** |
| Secondary hospitals | -0.270 (-0.356,-0.184)*** | 0.254 (-1.517,2.025) | 0.472 (0.238,0.706)*** | 0.016 (-2.207,2.238) | 0.058 (-0.0181,0.297) | 2.67190. 321,5.020)* | 0.532 (0.361,0.708)*** | 19.707 (18.246,21.167)*** |
| Primary hospitals | -0.160 (-0.251,-0.070)** | 0.428 (-1.003,1.859) | 0.177 (-0.011,0.365) | 0.453 (-1.126,2.032) | 0.189 (0.024,0.354)* | 0.925 (-0.920,2.770) | 0.373 (0.234,0.511)*** | 13.465 (12.204,14.727)*** |
| **Length of stay** | | | | | | | | |
| Total hospitals | -0.076 (-0.122,-0.030)** | -0.061 (-0.639,0.517) | 0.071 (0.011,0.132)* | 0.228 (-0.131,0.588) | -0.019 (-0.059,0.021) | 0.114 (-0.546,0.774) | 0.053 (0.004,0.101)* | 10.619 (9.869,11.369) |
| Tertiary hospitals | -0.137 (-0.187,-0.087)*** | -0.442 (-1.023,0.138) | 0.210 (0.123,0.296)*** | -0.187 (-1.036,0.662) | -0.107 (-0.180,-0.033)** | 0.255 (-0.641,1.150) | 0.103 (0.039,0.167)** | 13.079 (12.228,13.930)*** |
| Secondary hospitals | -0.140 (-0.225,-0.054)** | 0.077 (-0.811,0.964) | 0.122 (0.020,0.223)* | 0.230 (-0.272,0.732) | 0.010 (-0.045,0.066) | -0.331 (-1.655,0.993) | 0.123 (0.019,0.227)* | 12.989 (11.596,14.381) |
| Primary hospitals | -0.013 (-0.035,0.009) | -0.260 (-0.703,0.184) | 0.010 (-0.038,0.058) | 0.416 (0.086,0.747)* | -0.016 (-0.062,0.030) | -0.009 (-0.548,0.529) | -0.008 (-0.050,0.034) | 8.291 (7.917,8.665)*** |

Note: * $p < 0.05$; ** $p < 0.01$; *** $p < 0.001$.

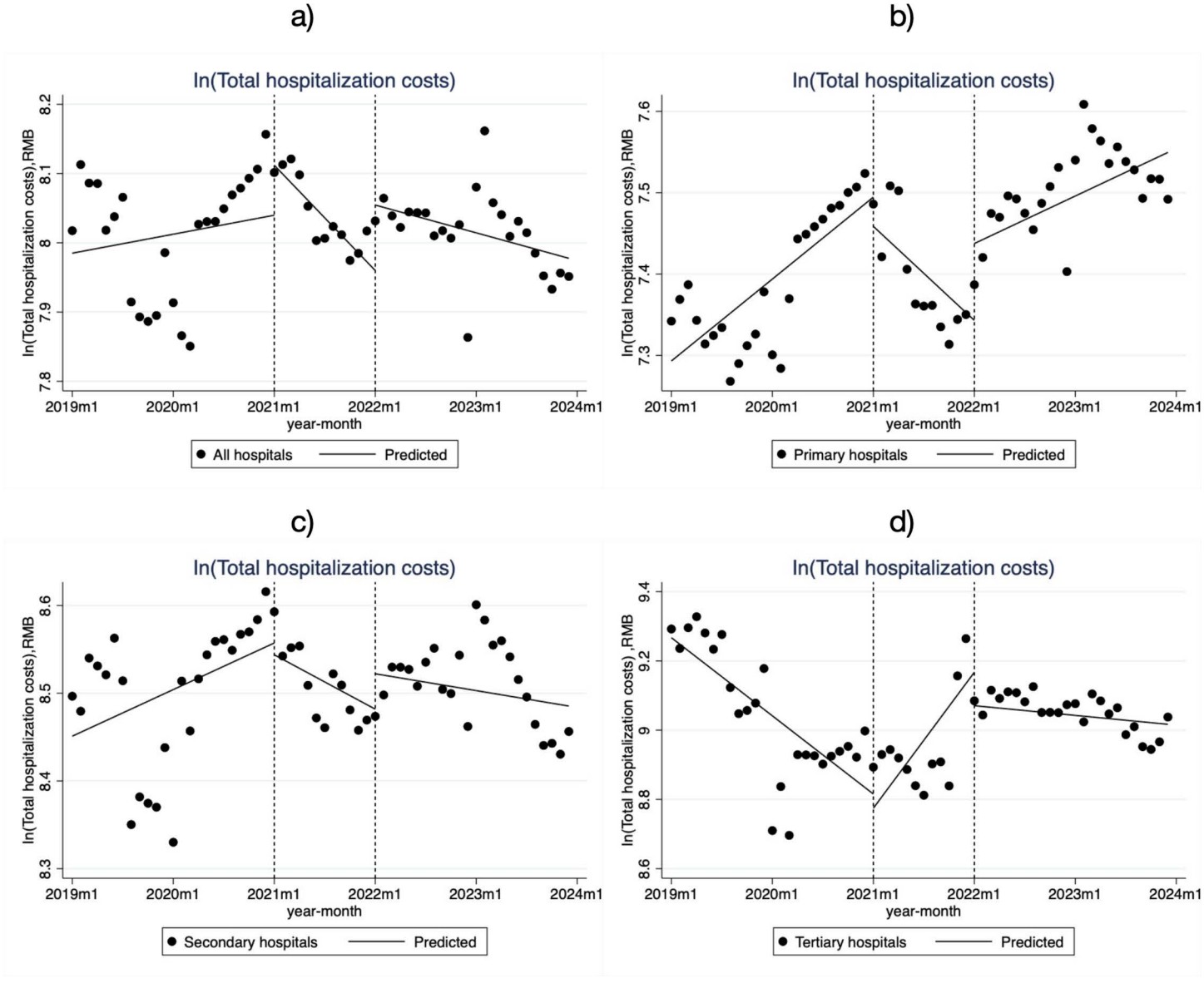

**Fig 1. Monthly trend of total hospitalization cost per low-income inpatient case in City S. a)** all hospitals; **b)** Primary hospitals; **c)** Secondary hospitals; **d)** Tertiary hospitals.

## Discussion

This study focuses on the direct impact of DIP reforms on inpatient services for low-income populations, a design choice driven by two key considerations. First, the DIP payment system currently primarily covers inpatient services, and outpatient reimbursement policies have not yet been integrated with the inpatient system. As a result, inpatient behaviors serve as the optimal entry point for observing the short-term effects of the policy. Second, low-income populations are significantly more sensitive to the cost of inpatient services than to outpatient services. Therefore, improvements in the economic accessibility of inpatient care are more directly relevant to enhancing their health equity. While the lack of outpatient

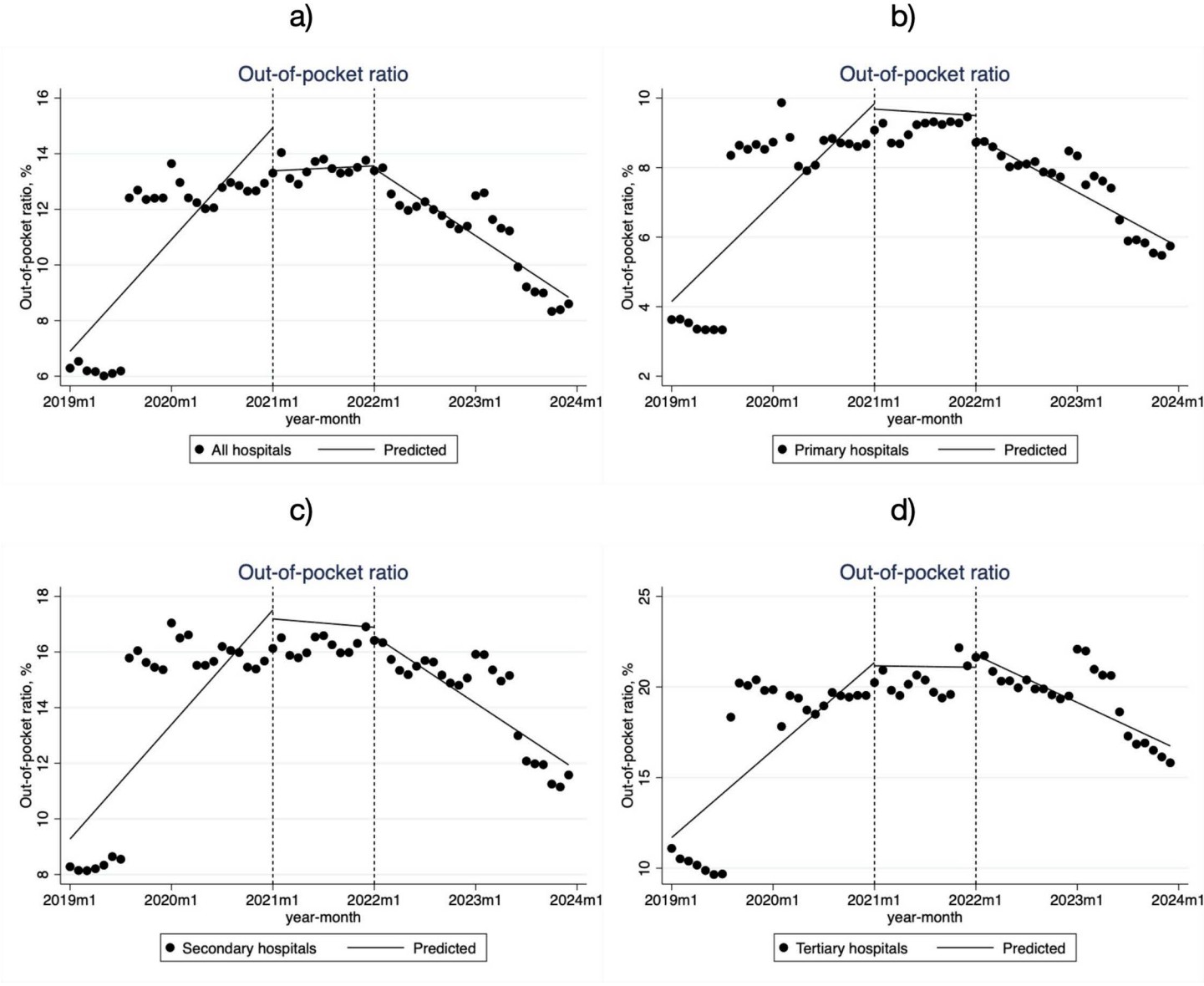

**Fig 2. Monthly trends in OOP ratios for low-income inpatients in City S. a)** all hospitals; **b)** Primary hospitals; **c)** Secondary hospitals; **d)** Tertiary hospitals.

data may limit a comprehensive assessment of the system's cascading effects, the heterogeneous changes observed in inpatient behavior provide crucial insights into the micro-mechanisms underlying payment reform.

To our knowledge, this is the first empirical study to assess the impact of DIP payment reforms on inpatient services for low-income populations across different hospital levels in pilot cities in central China. We used Medicare claims data from City S for this study. Given the year-long policy pilot period, a multi-stage interrupted time-series analysis was employed to assess the impact of DIP payment reforms on the low-income population's exposure to hospitalization at different hospital tiers. The findings indicate that, overall, DIP payment reforms have contributed to controlling healthcare costs and

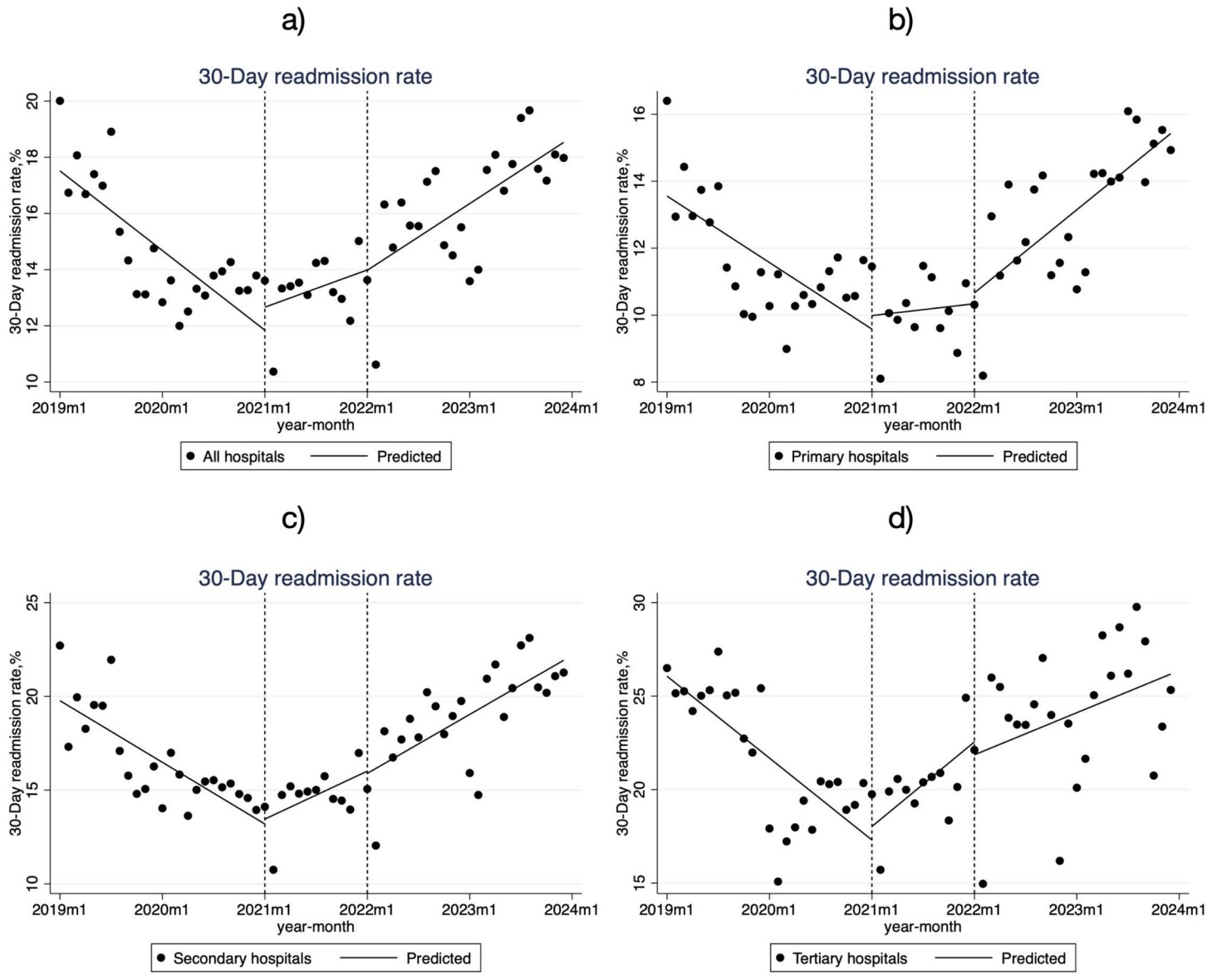

**Fig 3. Monthly Trends in 30-Day Readmission Rates for Low-Income Inpatients in City S. a)** all hospitals; b) Primary hospitals; **c)** Secondary hospitals; **d)** Tertiary hospitals.

improving efficiency, consistent with results from other middle-income regions. However, there are systematic differences in how the policy affects hospitals of varying tiers. For primary hospitals, the policy has led to some benefits, but also triggered moral hazard; secondary hospitals face more prominent structural conflicts, with a balancing act between cost control and quality; and tertiary hospitals have effectively addressed the rigid medical needs of low-income critically ill patients while successfully controlling hospitalization costs. To better understand the underlying reasons, we analyze the different stages of the reform separately.

In the pre-reform period, hospitals operated under a FFS model, which incentivized over-treatment to maximize revenue [34,35]. Tertiary hospitals, with higher costs and rising OOP ratios, saw significant declines in total hospitalization

**Fig 4. Monthly Trends in LOS for Low-Income Inpatients in City S. a)** all hospitals; **b)** Primary hospitals; **c)** Secondary hospitals; **d)** Tertiary hospitals.

costs, likely due to low-income patients' sensitivity to price, which led them to forego non-essential treatments to reduce financial burdens [36]. This claim is supported by previous studies, which have concluded that when the health insurance deductible increases by 10%, health care costs also decrease by 1% to 2% [37].In contrast, primary hospitals, with simpler and more affordable services, maintained relatively stable costs during this period.

During the DIP payment reform transition, the new methodology was implemented in a limited number of medical institutions in S City, with payment standards still being refined. Our study found that hospitalization costs in tertiary hospitals initially decreased but later increased, while costs in primary and secondary hospitals followed the opposite trend. This

differs from findings in southern and western prefecture-level cities and may be due to differences in the target population and implementation strategies [24,27].This phenomenon could be attributed to changes in the types of patients treated at tertiary hospitals, discrepancies in the payment mechanisms, and hospitals' adaptive adjustments. Low-income patients in tertiary hospitals tend to have more complex conditions, especially chronic diseases, which are reflected in the longer hospitalization periods. Similar patterns were seen in a northern prefecture-level city, where the DIP model promoted hierarchical treatment, causing more serious cases to shift to tertiary hospitals [26]. As the policy was still being refined, an increase in high-cost patients likely drove up overall costs in tertiary hospitals.Moreover, the DIP payment model was not fully implemented during the transition period, leaving some high-cost diseases inadequately covered or under-reimbursed. This resulted in higher out-of-pocket expenses for low-income patients. Hospitals may have also shifted costs to patients or adjusted treatments due to financial pressures. Studies on DRGs have shown that low reimbursement rates for high-severity cases often lead hospitals to shift costs to patients or adjust care, and the early stages of the DIP reform likely compounded these challenges [38].

In the advancement stage, as the DIP payment reforms comprehensively covered all hospitals, cost control improved, especially in secondary and tertiary hospitals. These hospitals, leveraging better resources and management systems, optimized diagnostic pathways and resource allocation to avoid exceeding payment standards, leading to steady declines in hospitalization costs [24,27]. Our results showed that the proportion of patients attending primary hospitals gradually increased during the policy advancement period, reaching 57.14% in 2023. This may be due to the focus on primary care for common illnesses and the use of price incentives. Patients often seek care at primary hospitals to benefit from higher reimbursement rates. The rise in total hospitalization costs for patients in primary hospitals reflects two opposing dynamics: while the policy promotes hierarchical treatment, it also suggests that policymakers have not adequately allocated resources to primary healthcare institutions to support patient diversion. As a result, healthcare costs may rise before they are controlled, as primary hospitals lack the necessary resources and management capacity. When payment policies require both cost control and basic protection, primary care hospitals may face the dilemma of 'doing more while losing more.'

Our study found a general increase in 30-day readmission rates for low-income patients across different hospital tiers following the implementation of DIP payment reform, which could suggest a decline in the quality of care. However, patients from lower economic backgrounds have a significantly higher risk of readmission compared to those from more affluent groups [39]. Hospitals serving disadvantaged populations also face penalties for higher readmission rates, making it unlikely that quality decline alone explains the observed increase in readmissions [40]. To better understand the changes in readmission rates, we will further investigate the circumstances faced by low-income patients at different hospital levels, specifically distinguishing between justified readmissions, potential overuse of healthcare services, and fragmented hospitalizations.

Primary and secondary hospitals primarily serve patients with chronic and common conditions, and DIP reforms have lowered out-of-pocket hospitalization costs, easing the financial burden on low-income patients. While this reduction in financial pressure improves healthcare accessibility and meets patient needs, it also introduces potential problems. The simultaneous rise in hospitalization costs and readmission rates at primary hospitals reveals a threshold sensitivity effect: when out-of-pocket costs fall below a critical level (≤5% in this study), patients' cost sensitivity diminishes, leading to an increased demand for non-acute hospitalizations [41,42]. This phenomenon, observed in other studies on low-income populations, aligns with the moral hazard theory proposed by Arrow, where individuals may overuse healthcare resources when costs are low or partially covered [43–45]. This may be due to the DIP payment model covering only inpatient services, meaning that patients do not receive higher reimbursement rates for outpatient visits [44].Additionally, the increase in readmission rates at primary hospitals could be due to insufficient treatment quality or inadequate resource allocation [46,47]. To address this issue, S City could implement an "outpatient-inpatient reimbursement linkage," creating a lump-sum payment package for chronic disease management. This system could provide additional reimbursement when

patients comply with outpatient follow-up protocols, incentivizing preventive care. Strengthening outpatient reimbursement rates and coverage would ensure that patients receive adequate financial support for outpatient care, thus reducing unnecessary hospitalizations. Furthermore, improving the service capacity of primary care facilities by investing in medical equipment, pharmaceutical supplies, and human resources is crucial to ensuring high-quality chronic disease management. Finally, setting clear hospitalization thresholds and auditing mechanisms could ensure that hospital admissions are medically necessary, helping prevent excessive hospitalizations driven by financial incentives.

Our finding that secondary hospitals adopt a "shorter stay + more frequent" strategy to balance cost-control pressures with revenue maximization offers new insights into the strategic behaviors of middle-tier hospitals. Specifically, this behavior represents an efficiency-quality trade-off, which differs fundamentally from the "fragmented hospitalization" strategies observed in developed countries [10,48]. Disaggregated hospitalization involves hospitals splitting a single treatment into multiple admissions to circumvent DRG payment limits, thus increasing health insurance payments by artificially inflating service volume. In contrast, secondary hospitals in China face not only the pressures of payment reforms but also the responsibility of patient referrals [49]. This suggests that their strategy is not about merely increasing hospitalizations, but about achieving cost containment by shortening lengths of stay under the DIP cost-control framework. However, this approach may lead to inadequate care or incomplete treatment, resulting in patients needing to be readmitted for recurring conditions shortly after discharge. This reflects a typical crowding-out effect, where short-term cost-control measures undermine the quality of care.These findings highlight a key issue for policymakers: in the DIP payment model, relying solely on cost-control indicators may lead to unintended consequences, such as reduced care quality. To mitigate this, quality assurance mechanisms need to be integrated into the payment system, such as incorporating readmission rates and patient satisfaction into the factors that adjust reimbursement rates. Furthermore, the unique role of secondary hospitals as referral centers should be specifically compensated to avoid financial imbalances caused by their public service obligations.

Compared to primary and secondary hospitals, tertiary hospitals treat patients with more complex medical conditions, higher hospitalization costs, and greater out-of-pocket expenses, which may strain their living costs. For patients with lower insurance coverage, their healthcare demand is often suppressed, but when insurance reduces the cost of medical services or increases reimbursement rates, these patients are more likely to seek care. This suggests that the demand for critical care is somewhat rigid and cannot be entirely explained by moral hazard [50,51]. Tertiary hospitals in S city mainly treat critically ill patients, such as those with malignant tumors or immune system diseases. These conditions require frequent, intensive treatments like chemotherapy and immunotherapy, making hospitalization a necessity for disease management. For patients undergoing multiple treatment cycles, the DIP reform reduces the cost of each hospitalization, but complete treatment often requires several admissions.The observed increase in readmission rates, in this case, reflects an improvement in treatment completeness rather than a decline in service quality, differentiating it from unnecessary hospitalizations driven by cost perception issues in tertiary hospitals. However, the possibility remains that tertiary hospitals may be reducing hospitalization costs and days by disaggregating treatments. While increased hospitalization rates may indicate better medical support, the challenge remains for healthcare providers and policymakers to balance the needs of critically ill patients with controlling healthcare expenditures. To address this, a bundled payment mechanism for treatment cycles could be implemented for tertiary hospitals. For instance, combining billing for planned readmissions within six months could help avoid misclassifying necessary readmissions as excessive treatment. Additionally, monitoring whether reduced inpatient stays breach clinical safety thresholds could serve as a strategy to ensure both cost control and patient care quality.

Although this study focused on pilot cities in central China, its findings align closely with empirical results from regions with similar economic levels. For example, the DIP reform in Tai'an City led to a decrease in the proportion of patients treated in tertiary and secondary hospitals, while the proportion of patients in primary hospitals increased [26]. At the same time, the proportion of complex cases (PRCP) and the case-mix index (CMI) significantly rose in tertiary hospitals,

while primary care organizations struggled to meet the increased demand. This study similarly observes a temporary rise in hospitalization costs at tertiary hospitals, alongside an increase in the severity of patient conditions, while primary care facilities faced challenges in addressing the growing demand. This cross-regional consistency suggests that variations in hospital service capacity, rather than regional economic differences, are the primary drivers of heterogeneity in the effects of the DIP reform. Additionally, the distribution of low-income patients across hospital levels in this study contrasts with findings for the broader population in Tai'an, offering fresh insights into disadvantaged groups and providing valuable evidence for the design of targeted policies.

## Limitations

Although this study focused on low-income populations and analyzed inpatient data before and after the DIP reform, it did not include outpatient or general practice data, potentially underestimating the reform's overall impact on healthcare services for this vulnerable group. Future research should incorporate outpatient data and patient health outcomes to provide a more comprehensive understanding of how the policy affects healthcare utilization among low-income populations. Additionally, as the study was limited to one pilot city, the external validity of the results remains unverified in other regions. Cross-regional analyses are needed to examine the generalizability and adaptability of the DIP payment reform across different healthcare settings and demographic groups. While the findings highlight policy effects for low-income patients, the absence of qualitative analysis on hospital and patient behavioral changes limits the understanding of the mechanisms driving these effects. Future studies should combine qualitative interviews and behavioral experiments to explore how hospital administrators and low-income patients respond to the policy's financial incentives and its broader implications.

## Conclusions

This study assessed the impact of DIP payment reforms on inpatient services for low-income populations across different hospital levels in a pilot city in central China. Using a multi-stage interrupted time-series analysis, the findings reveal that DIP reforms significantly influenced total hospitalization costs and 30-day readmission rates, with variations observed across hospital levels and reform stages. The reforms effectively reduced the financial burden on low-income patients, particularly by lowering hospitalization costs in secondary and tertiary hospitals during the advancement phase. However, increased costs in primary hospitals and rising readmission rates highlight unintended consequences, such as overutilization of inpatient services and potential inefficiencies in resource allocation.The dual impact of the DIP reforms underscores the importance of balancing cost control with equitable access to healthcare. While the policy successfully improved accessibility for low-income patients, challenges remain in mitigating moral hazard and ensuring optimal hospital practices. Policymakers should focus on refining reimbursement mechanisms, strengthening regulatory frameworks, and encouraging outpatient and community-based care to reduce unnecessary hospitalizations.

## Supporting information

**S1 Table. Robustness tests.**
(DOCX)

## Author contributions

**Conceptualization:** Yingying Tao, Yile Li, Chengcheng Li, Xuehui Meng.

**Data curation:** Keyi Shen, Ziqian Jin, Dan Wu, Xuehui Meng.

**Formal analysis:** Keyi Shen, Yingying Tao, Yile Li.

**Methodology:** Keyi Shen, Chengcheng Li, Dan Wu, Xuehui Meng.

**Software:** Keyi Shen, Yingying Tao.

**Supervision:** Ziqian Jin, Dan Wu, Xuehui Meng.

**Visualization:** Chengcheng Li.

**Writing – original draft:** Keyi Shen, Yingying Tao, Yile Li.

**Writing – review & editing:** Ziqian Jin, Chengcheng Li, Xuehui Meng.

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
