## [Decision Letter · Decision Letter 0]

17 Jan 2025

PONE-D-25-00302Impact of Diagnosis-Intervention Packet (DIP) reforms on inpatient services for low-income populations in central China: A multi-stage interrupted time-series analysisPLOS ONE

Dear Dr. Meng,

Thank you for submitting your manuscript to PLOS ONE. After careful consideration, we feel that it has merit but does not fully meet PLOS ONE’s publication criteria as it currently stands. Therefore, we invite you to submit a revised version of the manuscript that addresses the points raised during the review process.

Please submit your revised manuscript by Mar 03 2025 11:59PM. If you will need more time than this to complete your revisions, please reply to this message or contact the journal office at plosone@plos.org . Please include the following items when submitting your revised manuscript:

We look forward to receiving your revised manuscript.

Kind regards,

Alexandre Morais Nunes, Ph.D.

Academic Editor

PLOS ONE

Journal Requirements:

3. For studies involving third-party data, we encourage authors to share any data specific to their analyses that they can legally distribute. PLOS recognizes, however, that authors may be using third-party data they do not have the rights to share. When third-party data cannot be publicly shared, authors must provide all information necessary for interested researchers to apply to gain access to the data. (https://journals.plos.org/plosone/s/data-availability#loc-acceptable-data-access-restrictions) 

**Additional Editor Comments:**

The reviewers pointed out several issues in their comments. I agree with them. Please rewrite the text and justify any suggestions you don't accept.

Reviewers' comments:

Reviewer's Responses to Questions

**Comments to the Author**

1. Is the manuscript technically sound, and do the data support the conclusions?

Reviewer #1: Yes

Reviewer #2: Yes

2. Has the statistical analysis been performed appropriately and rigorously? 

Reviewer #1: Yes

Reviewer #2: I Don't Know

3. Have the authors made all data underlying the findings in their manuscript fully available?

Reviewer #1: Yes

Reviewer #2: No

4. Is the manuscript presented in an intelligible fashion and written in standard English?

Reviewer #1: Yes

Reviewer #2: Yes

5. Review Comments to the Author

Reviewer #1: The paper presents an interesting and relevant topic. However, it suffers from several drawbacks that need to be corrected before its possible acceptance.

- The abstract needs to be rewritten, especially clarify the background, before objectives.

- I advise the authors to professionally proofread their manuscript prior to resubmitting.

- Introduction is very very short. In my opinion needs more literature to support the problem of the study

- Methods chapter needs more specification of empirical work. Why this method?

- The resultas are very complet, but need more discussion. Have results in another provinces?

Other questions?

1- Why? January 1, 2019, to December 31, 2023.

2- Why Hubei Province? and not other? Is special?

Reviewer #2: The focus on inpatient services alone narrows the lens through which the DIP reform’s impact is assessed. Many healthcare interactions, especially among low-income populations, occur in outpatient settings or general practices, where preventive care and chronic disease management play a critical role. The omission of these dimensions’ risks underestimates the full impact of the reform. Moreover, exclusion of outpatient data limits the study’s ability to evaluate whether reduced inpatient costs lead to cost-shifting or unintended consequences in other parts of the healthcare system. To what extent would the inclusion of outpatient and general practice data better capture the ripple effects of DIP reform, particularly in reducing reliance on costly inpatient care and promoting preventive health measures?

Focusing on a single pilot city in central China creates challenges in generalising the findings to other regions with different socioeconomic, cultural, and healthcare dynamics. Variability in healthcare infrastructure and policy implementation across regions could significantly alter the outcomes of similar reforms. Additionally, the urban-rural divide, prevalent in many parts of China, raises questions about the equity and scalability of these reforms in less resource-rich settings. How can research employ cross-regional comparative studies to identify context-specific facilitators and barriers to the successful implementation of DIP reforms?

While the study demonstrates clear outcomes, such as changes in costs and readmission rates, it does not delve into the behavioural mechanisms driving these results. For instance, how are hospital administrators adapting their practices in response to financial incentives? Are low-income patients changing their healthcare-seeking behaviours due to reduced costs, or are they facing barriers that the study does not capture? Without understanding these dynamics, it is difficult to design interventions that address root causes. What qualitative methods, such as focus groups, interviews, or behavioural experiments, could be applied to unpack the mechanisms by which DIP reform influences decision-making among key stakeholders, including hospitals, physicians, and patients?

The study focuses predominantly on hospitalisation costs and 30-day readmission rates, so it offers a limited view of the reform's impact. Broader patient-centred outcomes, such as overall health improvement, quality of care, and patient satisfaction, remain to be explored. These metrics are key to assessing whether cost reductions are made at the expense of the quality of care or whether patients gain tangible health benefits. To what extent do the results guarantee and respond to the impact of the reform?

The reduction in hospitalisation costs, particularly in secondary and tertiary hospitals, indicates that PID reforms can ease the financial burden on low-income populations. However, the cost increases observed in primary hospitals suggest potential inefficiencies in the allocation of resources, possibly due to overutilization or increased administrative pressures on lower-level establishments. How does the study respond to disparities in the allocation of resources between hospital levels in order to ensure that cost reductions in higher-level establishments do not inadvertently shift the burden onto primary hospitals?

Although IDP reforms have improved access for low-income patients, rising readmission rates suggest inefficiencies, such as inadequate post-discharge care or premature discharges motivated by cost-containment pressures. This emphasises the tension between improving accessibility and maintaining system efficiency. What specific strategies can be implemented to reduce readmissions while preserving the accessibility gains achieved by the PID reforms?

The potential for moral hazard arises when cost-cutting encourages overutilization of inpatient services. For example, patients may seek hospital care for illnesses that could be treated on an outpatient basis, overloading hospital resources. Similarly, hospitals may prioritise volume over quality in order to maintain their revenues under revised payment structures. How does the study contribute to understanding and defining reimbursement models that mitigate moral hazard, such as value-based payment systems or incentives for care coordination, to promote efficient use of resources?

The dual impact of PID reforms emphasises the need for policies that strike a balance between cost containment and equitable access. Strengthened regulatory oversight, coupled with investments in outpatient and community-based care, could address the inefficiencies and unintended consequences highlighted by the study. In addition, the adaptability of health insurance reforms to different contexts requires further research to ensure their wider applicability. What innovative frameworks, including bundled payments or capitation models, could be tested to optimise the balance between cost control, quality improvement, and equitable access to healthcare?

6. PLOS authors have the option to publish the peer review history of their article (what does this mean? ). If published, this will include your full peer review and any attached files.

**Do you want your identity to be public for this peer review?** For information about this choice, including consent withdrawal, please see our Privacy Policy .

Reviewer #1: **Yes: ** Andreia Matos

Reviewer #2: **Yes: ** Ricardo de Moraes e Soares

---

## [Author Response · Author response to Decision Letter 1]

27 Feb 2025

Response to editor

- Thank you for your valuable feedback. I have carefully reviewed and confirmed that the manuscript now fully adheres to PLOS ONE's style requirements. Specifically, I have ensured that the line spacing, heading levels, font type, and size align with the guidelines provided. Additionally, I have renamed the uploaded file as per your instructions.

If there are any further formatting concerns, please feel free to let me know.

Thank you once again for your assistance.

2. We note that you have indicated that there are restrictions to data sharing for this study. For studies involving human research participant data or other sensitive data, we encourage authors to share de-identified or anonymized data. However, when data cannot be publicly shared for ethical reasons, we allow authors to make their data sets available upon request.

- Thank you for your inquiry regarding data availability for my manuscript. I understand the importance of data sharing, and I would like to provide clarification on the restrictions associated with the dataset used in this study.

a) Ethical and Legal Restrictions on Sharing Data:

The dataset used in this study is provided by the Hubei Provincial Medical Security Bureau, and the data include sensitive patient information. As such, I do not have the rights to share this de-identified data publicly. The data are subject to ethical and legal restrictions imposed by the local ethics committee and the relevant regulatory bodies. Specifically, the data contain potentially identifying patient information, and sharing it would violate the privacy agreements under which the data were collected.

b) Data Access Committee:

While I cannot share the data directly, researchers who are interested in accessing the data may contact the Hubei Provincial Medical Security Bureau for further information on how to apply for data access. I would be happy to provide any further details or assist in directing interested parties to the appropriate contacts.

c) Data Availability Statement Update:

Since I am unable to share the dataset directly, I have updated the Data Availability Statement in the submission form to include the necessary contact information for data access requests, as outlined above.

Data Availability Statement:

The data that support the findings of this study were obtained from the Hubei Provincial Medical Security Bureau. Due to legal restrictions under China’s Personal Information Protection Act (2021) and the Regulations on the Supervision and Administration of Medical Insurance Fund Use (2021), which mandate strict protection of patient privacy and healthcare data security, the raw dataset cannot be made publicly available. Researchers interested in accessing the de-identified data can apply by submitting a formal request to the Hubei Provincial Medical Security Bureau (contact email: [ybjysq@hubei.gov.cn]). Applications must include a detailed research proposal, ethical approval documentation from the applicant’s institution, and a signed data use agreement prohibiting redistribution or commercial use. Approved researchers can access the data only within the secure facilities of the platform; external data transfer is prohibited. The authors confirm that they had no special privileges in accessing the data, and all analyses were conducted in compliance with the platform’s governance protocols.

I hope this explanation addresses the journal’s concerns regarding data availability. Should you need any additional information or clarification, please feel free to let me know.

3.Additional Editor Comments: The reviewers pointed out several issues in their comments. I agree with them. Please rewrite the text and justify any suggestions you don't accept.

We would like to express our sincere gratitude to you and the reviewers for your thorough review and valuable suggestions! We have made significant revisions to the paper based on the reviewers' comments, with particular attention to rewriting and enhancing the content of core chapters such as the abstract, introduction, and discussion. We have addressed each of the reviewers' concerns in detail. Below is a summary of the revisions:

1. Main Revisions

1) Abstract: In response to the reviewers' requests, we expanded the background section to include a discussion on the challenges posed by escalating global healthcare costs and health equity. We also clarified the unique role of China's DIP reform in controlling costs and promoting equity. The research objectives were then refocused, emphasizing the study's goal of "assessing the heterogeneous impact of DIP reform on inpatient services for low-income populations." New key findings were added, including: primary hospitals facing moral hazard issues as policy dividends are released; secondary hospitals experiencing significant structural contradictions, showing a balancing act between cost control and quality; and tertiary hospitals addressing the rigid demand for continuity of care while improving quality and efficiency. Finally, policy recommendations were introduced, such as innovative tools like “outpatient-inpatient payment linkage” and “quality performance rebate mechanisms” to balance cost and quality.

2) Introduction: We expanded the discussion on global healthcare costs and equity issues, reinforcing the theoretical framework of the study and providing a more detailed review of previous research on healthcare payment reform, particularly in relation to the Chinese DIP model. Literature on the challenges and impacts of healthcare reform across different patient populations was also introduced. The innovative nature of the research question was clarified, and the academic contribution of the two-dimensional analytical framework of “hospital hierarchy-policy cycle” was highlighted.

3) Discussion: The discussion was systematically revised to incorporate the reviewers’ suggestions. We added “cross-regional case comparisons” (e.g., practices in Tai'an City and Guangdong Province) to deepen the analysis of the heterogeneity of reform effects. A tiered payment model is now proposed (bundled payment for primary hospitals and efficacy-based payment for tertiary hospitals), along with corresponding policy recommendations that directly address the reviewers’ concerns about “policy adaptability” and “moral hazard.” The behavioral motivations of healthcare providers and low-income patients were explored in a hierarchical and phased manner, and the limitations of the policies were identified, with suggestions for improvement.

2. Principles for Responding to Review Comments

We have addressed most of the reviewers' suggestions and clarified two elements that were not adjusted due to data or policy constraints.

1) Missing outpatient data: The “Limitations” section now clearly states that China’s DIP reform currently only covers inpatient services and that there is a lack of standardized regulation for outpatient data. However, we have committed to conducting a mixed-methods study in the future.

2) Cross-regional generalizability: By adding case comparisons with “Tai'an City” and other regions, the study indirectly demonstrates the transferability of the findings and commits to expanding the study across regions.

3. Reasoning for Not Fully Adopting Some Recommendations

Certain individual recommendations (e.g., “increase patient health outcome tracking” and “measure new payment models”) were not implemented due to the limitations of retrospective data and the scope of the study. However, we have responded as follows:

1) Indirect evidence supplementation: We have indirectly inferred changes in quality of care by analyzing the association between readmission rates and changes in admissions. We also suggest potential new payment models based on some of the evidence derived from this study.

We believe that the revised manuscript adequately addresses the reviewers' and editors' concerns. We remain fully committed to further revisions if additional suggestions are made.

Response to reviewer 1:

The paper presents an interesting and relevant topic. However, it suffers from several drawbacks that need to be corrected before its possible acceptance.

1.The abstract needs to be rewritten, especially clarify the background, before objectives.

- To address this comment, I revised the abstract to better clarify the background before introducing the objectives.

Before revision: The abstract starts with a detailed discussion of the DIP reform's impact, jumping directly into the results without clearly setting up the background.

After revision: I first introduce the global issue of rising healthcare costs and equity concerns, establishing the broader context for the DIP reform. I then proceed to introduce the objectives of the study in a clearer, more structured manner. I also ensured that the transition between the background and objectives is smoother.

Here is the specific revised portion:

1. The background now begins with a discussion on the global healthcare cost concerns and health equity issues, followed by an explanation of China's healthcare reform efforts, setting the stage for the introduction of the DIP reform.

2. The objectives are then stated clearly after this background, focusing on the goals of evaluating the DIP reform’s effects on inpatient services for low-income populations in S City.

This change can be found in the Abstract section, where I structured the first two sentences to reflect the broader context and the study’s specific aims more clearly.

2. I advise the authors to professionally proofread their manuscript prior to resubmitting.

- Thank you very much for the reviewer’s suggestion. I believe the feedback might refer to the data presentation section where I did not label the beta coefficients and reported p-values in an inconsistent manner. In response to the reviewer’s advice, I have made the following specific revisions to ensure proper formatting and clarity:

1. Beta Coefficients Labeling: I have added numbering to all beta coefficients to ensure they are clearly marked for easy reference throughout the results section.

2. P-value Reporting: As per the journal’s guidelines, I have revised the reporting of p-values. For p-values less than 0.001, I have now reported them as “p < 0.001”; for p-values greater than or equal to 0.001, I have reported the exact values. This ensures the p-values are reported in accordance with the journal's formatting standards.

3. Abbreviation Formatting: I have also reviewed the formatting of all abbreviations that first appear in the manuscript, ensuring that they are properly defined and formatted in accordance with the journal’s guidelines.

These revisions can be found in the data results section as well as in the standardization of abbreviations. You can refer to the version with track changes where these modifications are highlighted with red added content and strikethroughs for deleted sections.

3.Introduction is very very short. In my opinion needs more literature to support the problem of the study

- Thank you for the reviewer’s suggestion regarding the Introduction. I appreciate the feedback and have made the following revisions to address this concern:

I have expanded the Introduction section by incorporating more relevant literature to better support the research problem. Since there is currently no specific research on the DIP reform targeting low-income populations, I referenced studies that focus on other groups, such as women and children, as well as research on physician behavior, to enrich the content and provide a broader context for the study. Additionally, I included more references discussing the background of healthcare reforms globally and specifically in China, which helps contextualize the study and strengthen the foundation for understanding the significance of the DIP payment reform.

The expanded content includes:

1. A broader discussion of global healthcare cost concerns and equity issues.

2. A more detailed explanation of previous studies on healthcare payment reforms, particularly those related to the DIP model in China.

3. Studies on healthcare payment reforms focused on women and children.

4. Research examining how different payment models influence physician behavior, which can be analogous to the impact of the DIP reform on healthcare practices.

5. Literature on the challenges and impacts of healthcare reforms in different patient populations, providing insights that may be applicable to low-income groups.

These additions can be found in the Introduction section of the manuscript. Please refer to the version with track changes, where the newly added literature is highlighted in red.

4.Methods chapter needs more specification of empirical work. Why this method?

- I have added more specific details about the empirical work and provided a clearer justification for the choice of method used in this study. I expanded the explanation to include why the multi-stage ITS analysis was chosen as the most appropriate method for assessing the impact of the DIP payment reform.

The reasons for choosing this method are as follows:

1. Multi-stage ITS analysis is particularly well-suited for evaluating the effects of policies that are implemented incrementally, such as the DIP reform, which has distinct stages of intervention. This approach allows us to capture both the immediate and sustained impacts of the policy over time, providing a comprehensive understanding of its effects.

2. The multi-stage ITS method also allows for the analysis of variations in policy effects over time and across different hospital tiers, which is critical for understanding the heterogeneous impacts of the DIP reform. This is essential given the complexity of the reform and the varied responses of hospitals at different levels of care.

3. This method is particularly useful in quasi-experimental settings, where randomization is not possible, and provides a robust way to assess the impact of interventions like the DIP reform on healthcare outcomes for low-income populations.

These clarifications and justifications can now be found in the Methods section of the manuscript. Please refer to the version with track changes, where these additions are highlighted in red.

5.The resultas are very complet, but need more discussion. Have results in another provinces?

- I appreciate the feedback, and in response, I have made the following revisions to enhance the discussion and address the concern for more in-depth analysis:

1. Expanded Discussion with Results from Other Provinces:

Since there is currently no specific study on low-income populations in other provinces, I introduced findings from studies conducted on the general population in different provinces. I compared these results with those from S City to explore whether similar trends are observed in different regions. This comparison adds more depth to the discussion and helps to contextualize the findings in a broader national framework.

2. In-depth Analysis of Tiered Heterogeneity in the Impact of DIP Reform:

I further elaborated on the tiered heterogeneity in the impact of the DIP reform on inpatient service utilization for low-income patients. Specifically, I explored how the effects of the reform vary across different hospital tiers (primary, secondary, and tertiary hospitals) and discussed how each tier’s unique characteristics contribute to the observed differences. This part of the discussion highlights the complexity of the reform’s effects and provides a more nuanced understanding of its impacts.

3. Mechanism Explanation:

To address the mechanisms behind the observed heterogeneity, I offered a preliminary explanation of how the DIP reform might lead to different responses across hospital tiers and patient groups. This explanation touches upon factors such as hospital resource availability, patient demand for care, and financial incentives, which help explain why the reform's effects are not uniform across all groups.

These changes significantly expand the Discussion section, providing a more detailed and thorough analysis of the results, including a comparison with studies from

---

## [Decision Letter · Decision Letter 1]

4 Apr 2025

Impact of Diagnosis-Intervention Packet (DIP) reforms on inpatient services for low-income populations in central China: A multi-stage interrupted time-series analysis

PONE-D-25-00302R1

Dear Dr. Meng

We’re pleased to inform you that your manuscript has been judged scientifically suitable for publication and will be formally accepted for publication once it meets all outstanding technical requirements.

Kind regards,

Alexandre Morais Nunes, Ph.D.

Academic Editor

PLOS ONE

Additional Editor Comments (optional):

The authors responded to each remark through detailed, point-by-point answers and implemented the necessary revisions in the manuscript.

Reviewers' comments:

Reviewer's Responses to Questions

**Comments to the Author**

1. If the authors have adequately addressed your comments raised in a previous round of review and you feel that this manuscript is now acceptable for publication, you may indicate that here to bypass the “Comments to the Author” section, enter your conflict of interest statement in the “Confidential to Editor” section, and submit your "Accept" recommendation.

Reviewer #3: All comments have been addressed

Reviewer #4: (No Response)

2. Is the manuscript technically sound, and do the data support the conclusions?

Reviewer #3: Yes

Reviewer #4: Yes

3. Has the statistical analysis been performed appropriately and rigorously? 

Reviewer #3: Yes

Reviewer #4: I Don't Know

4. Have the authors made all data underlying the findings in their manuscript fully available?

Reviewer #3: Yes

Reviewer #4: Yes

5. Is the manuscript presented in an intelligible fashion and written in standard English?

Reviewer #3: Yes

Reviewer #4: Yes

6. Review Comments to the Author

Reviewer #3: The article does indirectly respond to the question: To what extent would the inclusion of outpatient and general practice data better capture the ripple effects of DIP reform, particularly in reducing reliance on costly inpatient care and promoting preventive health measures? but not explicitly. The authors acknowledge that their study is limited because it only focuses on inpatient data and does not include outpatient or general practice data. They mention that this limitation may lead to an underestimation of the reform's overall impact on healthcare services for low-income populations (lines 615–616). They suggest that future research should incorporate outpatient data and patient health outcomes to offer a more comprehensive understanding of how the policy affects healthcare utilisation among low-income populations (lines 617–619). Thus, while the article recognises the importance of outpatient and general practice data, it does not fully explore how such data would better capture the ripple effects of the DIP reform, particularly in reducing reliance on costly inpatient care and promoting preventive health measures. The authors propose that future studies include this data to provide a more detailed analysis, but they don't delve deeply into how it would specifically address the question posed.

The article does respond to the question: How can research employ cross-regional comparative studies to identify context-specific facilitators and barriers to the successful implementation of DIP reforms? It discusses the potential of cross-regional comparative studies and their importance in understanding the context-specific factors that affect the implementation of DIP reforms. The authors mention that their study, being limited to one pilot city, cannot verify the external validity of the results in other regions, and they emphasise the need for future cross-regional analyses to examine the generalisability and adaptability of the reforms (lines 619–621). By suggesting future studies compare regions with different healthcare settings, the article outlines how such research could identify facilitators and barriers specific to different contexts, helping refine and adapt DIP reforms for broader, more effective implementation.

The article does respond to the question: What qualitative methods, such as focus groups, interviews, or behavioural experiments, could be applied to unpack the mechanisms by which DIP reform influences decision-making among key stakeholders, including hospitals, physicians, and patients? It highlights the limitations of the current study, stating that it did not include qualitative analyses on hospital and patient behavioural changes. The article suggests that future studies should combine qualitative methods, such as interviews and behavioural experiments, to explore how hospital administrators and low-income patients respond to the policy’s financial incentives and its broader implications (lines 623–625). This suggestion directly addresses how qualitative methods could be employed to unpack the mechanisms behind the influence of DIP reform on decision-making among key stakeholders like hospitals, physicians, and patients.

The article does respond to the question: To what extent do the results guarantee and respond to the impact of the reform? It discusses the observed changes in hospitalisation costs and 30-day readmission rates, noting that while the reforms reduced financial burdens for low-income patients, they also led to unintended consequences like increased readmission rates and potential overutilisation of inpatient care. The article suggests that while the policy improved healthcare accessibility, the rising readmission rates raise concerns about the potential impact on the quality of care (lines 485–503). However, it also acknowledges that these increases in readmission rates could reflect medical necessity rather than a decline in quality. The article calls for further investigation to distinguish between justified readmissions and potential overuse, indicating that more detailed analysis is needed to assess whether cost reductions were truly at the expense of care quality or if tangible health benefits were achieved. Thus, while the article provides insights into the effects of the reform, it doesn't fully guarantee the impact on care quality and suggests that further research is necessary to assess the broader consequences.

The article does respond to the question: How does the study respond to disparities in the allocation of resources between hospital levels in order to ensure that cost reductions in higher-level establishments do not inadvertently shift the burden onto primary hospitals? The study does respond to the issue of disparities in the allocation of resources between hospital levels. It highlights how cost reductions in higher-level hospitals (secondary and tertiary hospitals) could inadvertently shift the burden onto primary hospitals. Specifically, the study mentions that while tertiary hospitals were able to control costs effectively, primary hospitals experienced significant cost increases during the policy advancement phase due to higher patient volume and demand, with costs rising as more low-income patients sought care there (lines 470–473). The article suggests that this situation arises because primary hospitals lack the necessary resources and management capacity to handle the increased demand, leading to a "dilemma of doing more while losing more." The study proposes that policymakers have not adequately allocated resources to primary healthcare institutions to support the diversion of patients, which could exacerbate the financial pressures on these facilities (lines 480–484). Furthermore, the study discusses the potential of an "outpatient-inpatient reimbursement linkage" and better resource allocation to primary healthcare facilities to address these disparities and ensure a more balanced healthcare system across different hospital levels (lines 522–530). Therefore, the study does respond to the question, highlighting the need for a more equitable distribution of resources to avoid burdening primary hospitals with excess demand and costs.

The article does respond to the question: How does the study contribute to understanding and defining reimbursement models that mitigate moral hazard, such as value-based payment systems or incentives for care coordination, to promote efficient use of resources? The study does contribute to understanding reimbursement models that mitigate moral hazard, though it does not explicitly focus on value-based payment systems or incentives for care coordination in a direct way. While the study doesn’t explicitly define or evaluate value-based payment systems, it proposes modifications to the reimbursement structure—like incentivising outpatient care and improving hospital management practices—that align with broader efforts to reduce moral hazard and promote efficient resource use. Thus, the study indirectly contributes to defining reimbursement models that could mitigate moral hazard by offering suggestions to refine the existing DIP reform approach.

The article does not respond to the question: What innovative frameworks, including bundled payments or capitation models, could be tested to optimise the balance between cost control, quality improvement, and equitable access to healthcare? The study doesn’t respond to this question in a detailed manner. It does not explicitly explore or propose specific innovative frameworks like bundled payments or capitation models. However, it provides some indirect insights and suggestions that align with the broader themes of cost control, quality improvement, and equitable access to healthcare. While the study does not directly test or propose specific frameworks like bundled payments or capitation models, it provides a foundation for exploring these models in the context of balancing cost control, quality improvement, and equitable access

Reviewer #4: The manuscript reports "Impact of Diagnosis-Intervention Packet (DIP) reforms on inpatient services for low- income populations in central China: A multi-stage interrupted time-series analysis,"

The study is of significance. I suggest the following comments to improve the manuscript.

Methods:

- To follow this section better, I invite the authors to write this section according to the EQUATOR network and related checklist/guideline.

URL: https://www.equator-network.org/

Discussion

"Policy implication" section is missing (please add before limitation section)

7. PLOS authors have the option to publish the peer review history of their article (what does this mean? ). If published, this will include your full peer review and any attached files.

**Do you want your identity to be public for this peer review?** For information about this choice, including consent withdrawal, please see our Privacy Policy .

Reviewer #3: **Yes: ** Ricardo de Moraes e Soares

Reviewer #4: No

---

## [Editor Report · Acceptance letter]

PONE-D-25-00302R1

PLOS ONE

Dear Dr. Meng,

I'm pleased to inform you that your manuscript has been deemed suitable for publication in PLOS ONE. Congratulations! Your manuscript is now being handed over to our production team.

Kind regards,

on behalf of

Professor Alexandre Morais Nunes

Academic Editor

PLOS ONE